# Quantifying Tumor Heterogeneity via MRI Habitats to Characterize Microenvironmental Alterations in HER2+ Breast Cancer

**DOI:** 10.3390/cancers14071837

**Published:** 2022-04-06

**Authors:** Anum S. Kazerouni, David A. Hormuth, Tessa Davis, Meghan J. Bloom, Sarah Mounho, Gibraan Rahman, John Virostko, Thomas E. Yankeelov, Anna G. Sorace

**Affiliations:** 1Department of Radiology, The University of Washington, Seattle, WA 98104, USA; anumkaz@uw.edu; 2Oden Institute for Computational Engineering and Sciences, The University of Texas at Austin, Austin, TX 78712, USA; david.hormuth@austin.utexas.edu; 3Livestrong Cancer Institutes, The University of Texas at Austin, Austin, TX 78712, USA; jack.virostko@austin.utexas.edu; 4Department of Biomedical Engineering, The University of Texas at Austin, Austin, TX 78712, USA; tessa@austin.utexas.edu (T.D.); meghanjbloom@gmail.com (M.J.B.); sarahmounho@gmail.com (S.M.); grahman@utexas.edu (G.R.); 5Department of Diagnostic Medicine, The University of Texas at Austin, Austin, TX 78712, USA; 6Department of Oncology, The University of Texas at Austin, Austin, TX 78712, USA; 7Department of Imaging Physics, MD Anderson Cancer Center, The University of Texas, Houston, TX 77030, USA; 8Department of Biomedical Engineering, The University of Alabama at Birmingham, Birmingham, AL 35294, USA; 9Department of Radiology, The University of Alabama at Birmingham, Birmingham, AL 35294, USA; 10O’Neal Comprehensive Cancer Center, The University of Alabama at Birmingham, Birmingham, AL 35294, USA

**Keywords:** diffusion-weighted MRI, dynamic contrast-enhanced MRI, habitat imaging, immunofluorescence, immunohistochemistry, paclitaxel, trastuzumab, BT-474

## Abstract

**Simple Summary:**

Tumor heterogeneity influences tumor progression and response to therapy, introducing a significant challenge in the treatment of breast cancer. We employed magnetic resonance imaging (MRI) to characterize tumor heterogeneity over time in response to treatment in a mouse model of HER2+ breast cancer. In a two-part approach, we first used quantitative MRI to identify unique subregions of the tumor (i.e., “tumor habitats”, resolving intratumoral heterogeneity), then used the habitats to stratify tumors prior to treatment into two distinct “tumor imaging phenotypes” (resolving intertumoral heterogeneity). The tumor phenotypes exhibited differential response to treatments, suggesting that baseline phenotypes can predict therapy response. Additionally, there were significant correlations between the imaging habitats and histological measures of vascular maturation, hypoxia, and macrophage infiltration, lending ex vivo biological validation to the in vivo imaging habitats. Application of these techniques in the clinical setting could improve understanding of an individual patient’s tumor pathology and potential therapeutic sensitivity.

**Abstract:**

This study identifies physiological habitats using quantitative magnetic resonance imaging (MRI) to elucidate intertumoral differences and characterize microenvironmental response to targeted and cytotoxic therapy. BT-474 human epidermal growth factor receptor 2 (HER2+) breast tumors were imaged before and during treatment (trastuzumab, paclitaxel) with diffusion-weighted MRI and dynamic contrast-enhanced MRI to measure tumor cellularity and vascularity, respectively. Tumors were stained for anti-CD31, anti-ɑSMA, anti-CD45, anti-F4/80, anti-pimonidazole, and H&E. MRI data was clustered to identify and label each habitat in terms of vascularity and cellularity. Pre-treatment habitat composition was used stratify tumors into two “tumor imaging phenotypes” (Type 1, Type 2). Type 1 tumors showed significantly higher percent tumor volume of the high-vascularity high-cellularity (HV-HC) habitat compared to Type 2 tumors, and significantly lower volume of low-vascularity high-cellularity (LV-HC) and low-vascularity low-cellularity (LV-LC) habitats. Tumor phenotypes showed significant differences in treatment response, in both changes in tumor volume and physiological composition. Significant positive correlations were found between histological stains and tumor habitats. These findings suggest that the differential baseline imaging phenotypes can predict response to therapy. Specifically, the Type 1 phenotype indicates increased sensitivity to targeted or cytotoxic therapy compared to Type 2 tumors.

## 1. Introduction

Human epidermal growth factor receptor 2 (HER2) is overexpressed in approximately 15% of breast cancers within the United States [1]. HER2+ breast cancer is associated with more aggressive disease and poorer patient prognosis compared to HER2- breast cancer subtypes [2]. Treatment with trastuzumab, a humanized monoclonal antibody targeting the HER2 protein, has dramatically improved patient outcomes and reduced disease recurrence compared to traditional chemotherapeutics [3,4]. Unfortunately, trastuzumab elicits variable response within the HER2+ patient population, with overall response rates (i.e., the proportion of patients who have complete or partial response [5]) between 12–26% [2,6]. This variation in patient response to trastuzumab has in part been attributed to intratumoral heterogeneity, that is, the phenotypic and genotypic cellular diversity within a tumor. Intratumoral heterogeneity is known to influence tumor progression and introduces a significant challenge in the clinical treatment of HER2+ breast cancer, as diagnostically similar patients may respond differently to the same treatment strategy [7,8,9]. Previous efforts have demonstrated that heterogeneous HER2 expression exists within HER2+ tumors [10,11,12] and shown that HER2+ status alone does not necessarily correlate with objective response to HER2-targeted treatment [6,8,9,11]. The ability to quantitatively characterize intratumoral heterogeneity and its changes in response to therapy may provide clinically valuable information that can be used to guide treatment strategies.

Currently, in both clinical and preclinical settings, the standard techniques used to evaluate the biological characteristics of the tumor involve invasive procedures such as biopsies or tumor excision [7]. These techniques are susceptible to sampling error and may not provide an accurate description of the biological characteristics of the whole tumor [7,13]. Additionally, these methods preclude evaluation of the dynamics of the tumor microenvironment, such as vascular perfusion or metabolic activity, and only allow for measurement at a single time point. Conversely, quantitative medical imaging allows for noninvasive three-dimensional (3D) measurement of biological characteristics of the microenvironment throughout the entire volume of a tumor [14]. In particular, diffusion-weighted magnetic resonance imaging (DW-MRI) [15] and dynamic contrast-enhanced (DCE-) MRI [16] can quantitatively assess tissue cellularity and vascularity, respectively. Cellularity and vascularity are key tumor attributes which are altered by traditional chemotherapeutics such as paclitaxel as well as by targeted therapies, including trastuzumab. Accordingly, these MRI techniques have been shown to be predictive of breast cancer response in both the preclinical [17,18] and clinical [19,20] settings. Specifically, in the preclinical setting, DW-MRI has demonstrated utility as an early indicator of paclitaxel response based on measured decreases in tumor cellularity [21], and DCE-MRI has been utilized to measure trastuzumab-induced increases in vascular perfusion within HER2+ xenograft tumors [18]. 

The importance of the tumor microenvironment and its influence on cancer progression and therapeutic response is well-established [22,23]. Recent studies suggest new treatment strategies that manipulate the tumor microenvironment to increase the sensitivity of the tumor to subsequent therapy, thereby potentially minimizing drug resistance and metastasis [24,25]. Recently, multiparametric MRI data have been employed to spatially resolve intratumoral subregions, in an approach entitled habitat imaging [26]. Previously, we employed habitat imaging with multiparametric quantitative MRI to characterize intratumoral heterogeneity within two xenograft models of breast cancer [27]. The identified habitats were characterized by cellularity and vascularity metrics derived from imaging and biologically validated through correlation with corresponding habitats derived from histology data [27]. We demonstrated the utility of the habitat imaging approach for the characterization of temporal alterations in the tumor microenvironment, and identified changes in tumor composition (defined by habitats) associated with response to clinically-relevant therapeutics. 

In the present contribution, we employed multiparametric MRI data to identify tumor microenvironment habitats in a murine xenograft model of HER2+ breast cancer, then used the derived habitats observed at baseline (pre-treatment) to stratify tumors into two distinct “tumor imaging phenotypes”. This approach was used to resolve intertumoral differences during treatment with standard-of-care targeted and cytotoxic therapeutics. For each phenotype, the changes in overall tumor volume and percent tumor volume comprised of each habitat were measured in response to treatment of single-agent paclitaxel or trastuzumab therapy. Finally, at the study endpoint, we determined the relationship between tumor habitats and biological measures of cellularity and vascularity derived from immunohistochemistry and immunofluorescence data.

## 2. Materials and Methods

### 2.1. Cell Culture and Animal Model

BT-474 is a HER2+ human breast cancer cell line that has an established positive response to trastuzumab [28]. BT-474 cells (ATCC, Manassas, VA, USA) were cultured in improved minimum essential medium (IMEM, Corning, Tewksbury, MA, USA) with L-glutamine, 10% FBS, 20 ug/mL insulin, and 1% penicillin-streptomycin at 37 °C with 5% CO_2_. 

All animal procedures were approved by our institution’s animal care and use committee. Female athymic nude mice (The Jackson Laboratory, Bar Harbor, ME, USA) were implanted subcutaneously with a 0.72 mg 60-day release 17ß-estradiol pellet (Innovative Research of America, Sarasota, FL, USA); 24 h later, 1 × 10^7^ BT-474 breast cancer cells in serum-free media and 30% growth factor reduced Matrigel (Corning, Tewksbury, MA, USA) were injected subcutaneously into the flank of the mouse. Tumors (*N* = 86, Appendix A) were grown for 8–10 weeks until they reached approximately 235 mm^3^ in volume and were then randomly assigned to one of three treatment groups: control (saline), trastuzumab (10 mg/kg), or paclitaxel (10 mg/kg). Mice were treated with an intraperitoneal injection of drug or saline on days 0 (post-imaging) and 3 (Figure 1). Tumors that were larger than 350 mm^3^ at day 0 were excluded from the study (*N* = 2).

### 2.2. Longitudinal Tumor Growth Study

Longitudinal response to trastuzumab or paclitaxel was evaluated in a subset of mice (*N* = 24, Appendix A). Tumor volumes were measured with calipers once per week for 8–10 weeks prior to treatment, then three times per week following initiation of treatment. The percent change in tumor volumes was calculated on day 30 relative to the initiation of therapy.

**Figure 1 cancers-14-01837-f001:**
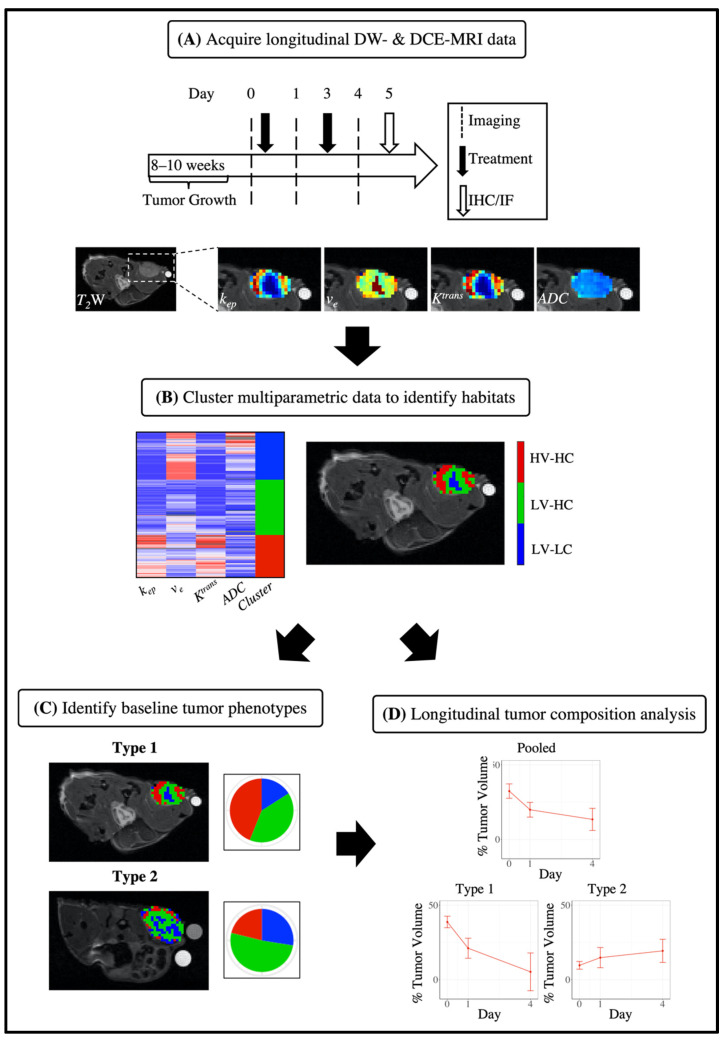
Pipeline for habitat imaging analysis and identification of tumor imaging phenotypes. Longitudinal DW- and DCE-MRI data was acquired on days 0 (pre-treatment), 1, and 4 and processed for each mouse to generate quantitative parameter maps (**A**, top row). Tumors were excised for immunohistochemistry (IHC) and immunofluorescence (IF) analysis at day 5. Multiparametric image data were clustered to identify tumor habitats (**B**, middle row). Baseline imaging data (day 0) of tumors were used to cluster tumors into two phenotypes based on tumor habitat composition of cellularity and vascularity (**C**, bottom row left). Tumors were separated by treatment group alone (pooled) or in addition to tumor imaging phenotype (Type 1, Type 2), and longitudinal analysis of tumor composition was analyzed in terms of the percent tumor volume each habitat comprised (**D**, bottom row right).

### 2.3. Magnetic Resonance Imaging

Tumor-bearing mice (*N* = 60, Appendix A) were imaged at baseline (day 0), prior to treatment, and then 24 h after each treatment dose (days 1 and 4, Figure 1); 24 h prior to baseline imaging, a jugular vein catheter was surgically implanted into mice for exogenous delivery of the contrast agent for DCE-MRI. Tumors were imaged using a 7 T preclinical MRI scanner (Bruker, Billerica, MA, USA) equipped with a 40 mm transmit–receive volume coil (Bruker, Billerica, MA, USA). The tumor region was located with a multi-slice localizer scan to establish the field of view (FOV), with the largest cross-section of the tumor located at the central slice. This FOV was used for the remainder of the imaging protocol. A volume selected around the tumor region was used to shim to optimize the homogeneity of the *B_0_* field around the tumor. High-resolution *T_2_*-weighted images were acquired over the entire tumor volume using a fast spin–echo pulse sequence with the following parameters: *TR* = 2500 ms, effective *TE* = 30.6 ms, NEX = 2, number of slices = 15, slice thickness = 1 mm, RARE factor = 8, no slice gap, and an acquisition matrix of 128 × 128 over a 34 × 34 mm^2^ FOV to yield a voxel size of 0.27 × 0.27 × 1 mm^3^. All other images were acquired over the same FOV and number of slices, but with an acquisition matrix of 64 × 64 over yielding a voxel size of 0.53 × 0.53 × 1 mm^3^.

Diffusion-weighted (DW) images were acquired using a standard pulsed gradient spin–echo sequence with three *b*-values (150, 500, and 800 s/mm^2^) and gradients applied simultaneously along three orthogonal directions (*x*, *y*, *z*). The image acquisition parameters were *TR*/*TE* = 2000/29.9 ms, number of acquisitions = 2, gradient duration *δ* = 3 ms, and gradient interval Δ = 20 ms. 

Pre-contrast *T_1_* maps were acquired using a segmented FLASH (segFLASH; segmented Fast Low Angle SHot) inversion recovery sequence. In the segFLASH sequence, after a 180° inversion pulse is applied, eighteen points of the *T_1_* relaxation curve are sampled with inversion times of 150 ms to 3550 ms with a 200 ms spacing. Additional segFLASH parameters were as follows: *α* = 15°, *TE* = 2.5 ms, number of acquisitions = 2, and number of segments = 8. Dynamic *T_1_*-weighted images were acquired using an RF-spoiled FLASH gradient–echo sequence with a temporal resolution of 10.8 s for 15 min with the imaging parameters *TR/TE* = 100/2.13 ms and *α* = 20°, NEX = 2. Pre-contrast images were acquired for approximately 2 min before a bolus of 0.05 mmol/kg Gd-DO3A-butrol (Gadovist, Bayer, Leverkusen, Germany) was delivered via a jugular catheter using an automated syringe pump (Harvard Apparatus, Holliston, MA, USA) at a rate of 2.4 mL/min. Additional details regarding animal setup, MRI acquisition, and MRI processing are provided in Appendix A.

### 2.4. Immunohistochemistry and Immunofluorescence

On day 5 (Figure 1), animals were intravenously injected with 60 mg/kg pimonidazole (Hypoxyprobe, Burlington, MA, USA) via the tail vein. One hour after pimonidazole injection, each animal was sacrificed and tumors were excised for biological analysis (*N* = 19, Appendix A). Tumors were cut in half at the longest cross-section corresponding to the central in vivo imaging plane, placed in optimal cutting temperature (OCT) compound, frozen, and stored at −80 °C. Frozen samples were sectioned, mounted onto glass slides, and fixed in 10% formalin prior to immunohistochemistry (IHC) and immunofluorescence (IF) staining. For IHC analysis, samples were stained with hematoxylin and eosin (H&E). For IF analysis, samples were stained with anti-CD31 (R&D Systems, Minneapolis, MN), anti-pimonidazole (Hypoxyprobe, Inc., Burlington, MA, USA), anti-ɑSMA, anti-CD45, or anti-F4/80 (Abcam, Cambridge, UK). Additional details regarding IF staining and image processing of IHC and IF data are provided in Appendix A.

### 2.5. Identification of MRI Tumor Habitats

Tumors with both DW- and DCE-MRI were used to spatially resolve **intra**tumoral heterogeneity through the identification of tumor habitats *(N* = 57, Appendix A). For each tumor ROI, multiparametric voxel data were extracted to yield a four-dimensional vector (*K^trans^*, *v_e_*, *k_ep_*, and *ADC*) for each voxel (Figure 1). All voxel data (i.e., all tumors at all time points) were then pooled and each parameter distribution was scaled to have a mean of zero and standard deviation of one, such that each parameter contributed equally to the clustering process. To identify tumor habitats, scaled voxel data were then clustered using an agglomerative clustering algorithm with the Ward linkage and Euclidean distance measure, as done in previous work [27]. The clustering process used no spatial information. Additional details on the process have been previously published [27]. The resultant dendrogram was cut with a *k* = 3 to identify three tumor habitats, based on previous findings [27] within the same tumor model.

For each automatically identified cluster, the mean value of each MRI parameter was calculated to determine the physiology a cluster (habitat) represented in terms of high or low “vascularity” and “cellularity”. Vascularity refers to the level of vascular perfusion and permeability within a voxel, quantified by *K^trans^* and *k_ep._* Cellularity refers to the cell density and cell membrane impermeability within a voxel, quantified by *ADC* and *v_e_*. Clusters with a mean *K^trans^ <* 0.2 min^−1^ or *k_ep_* < 0.5 min^−1^ were labeled as low-vascularity (LV) habitats, and otherwise were labeled as high-vascularity (HV) habitats. Clusters with a mean *v_e_ >* 0.6 or *ADC* > 7.5 × 10^−4^ mm^2^/s were labeled as low-cellularity (LC) habitats, and otherwise were labeled as high-cellularity (HC) habitats.

### 2.6. Discovery of Tumor Imaging Phenotypes 

To discern **inter**tumoral differences, we identified tumor imaging phenotypes as defined by the tumor habitats extracted from the pre-treatment quantitative MRI data (Figure 1). Each tumor was described by its baseline tumor composition using a 3D vector where each element consisted of the percent tumor volume comprised by each of the three identified tumor habitats at baseline. The tumor data were then pooled and each dimension scaled to obtain a mean of zero and a standard deviation of one. Baseline tumor data were then clustered using agglomerative clustering with the Ward linkage and Euclidean distance measure to identify phenotypes. The average silhouette method was used to determine the number of clusters (between two and ten) to cut the resultant dendrogram. 

### 2.7. Quantifying Longitudinal Alterations in Tumor Composition

The percent tumor volume comprised by each habitat was used to quantify the response of the tumor microenvironment to trastuzumab or paclitaxel therapy. The median percent tumor volume of each habitat was calculated for each treatment group. In a secondary analysis, tumors were subdivided by tumor imaging phenotype, then the median percent tumor volume of each habitat was calculated for each treatment within the phenotype subgroup. Tumor composition was evaluated in this manner for each imaging time point.

### 2.8. Statistical Analysis

A one-way analysis of variance (ANOVA) followed by Tukey’s honest significant difference test was used to test differences in mean parameter values between MRI-derived habitats as well as differences in percent change in tumor volume between treatment groups. A nonparametric Wilcoxon rank sum test was used to compare differences in tumor volume from baseline as well as differences between tumor imaging phenotypes; the Wilcoxon rank sum test was used to determine differences in percent tumor volume of each habitat from baseline during the course of therapy as well. Correlations were tested using Pearson’s product–moment correlation. In all statistical tests, a *p*-value less than 0.05 was considered significant; 95% confidence intervals are listed next to reported mean statistics and interquartile ranges are listed next to reported median statistics in parentheses. All statistical analyses were performed in R (R version 3.6.2, RStudio).

## 3. Results

### 3.1. Characterization of MRI Tumor Habitats

Figure 1 presents the study design for the discovery of tumor habitats from quantitative MRI data (resolving **intra**tumoral heterogeneity) and the identification of tumor imaging phenotypes using habitat information (resolving **inter**tumoral heterogeneity). Three habitats were identified from the clustering analysis of quantitative MRI data (Figure 2). To determine the physiology each habitat represented, we considered the mean values of *ADC* and *v_e_* to assess the cellularity and the mean values of *K^trans^* and *k_ep_* to assess the vascularity of each habitat. The clustering algorithm was able to identify physiologically distinct tumor habitats, as statistically significant differences between habitats were found for each parameter (Figure 2A). Figure 2B shows quantitative parameter maps for a representative tumor at baseline along with corresponding habitat maps. The first identified cluster had a mean *v_e_* of 0.83 ± 0.002 and mean *ADC* of 8.0 × 10^−4^ ± 3.5 × 10^−6^ mm^2^/s, and was thus labeled a “low-cellularity” (LC) habitat. Conversely, the second and third clusters had mean *v_e_* values of 0.48 ± 0.002 and 0.52 ± 0.002, respectively, and mean *ADC* values of 5.6 × 10^−4^ ± 1.3 × 10^−6^ and 6.1 × 10^−4^ ± 1.6 × 10^−6^ mm^2^/s, respectively. These habitats were labeled “high-cellularity” (HC) habitats. With respect to measures of vascularity, the third cluster had a mean *K^trans^* of 0.42 ± 0.002 min^−1^ and mean *k_ep_* 0.86 ± 0.006 min^−1^, and was thus labeled a “high-vascularity” (HV) habitat. The first and second clusters presented mean *K^trans^* values of 0.07 ± 0.001 and 0.10 ± 0.001 min^−1^ and mean *k_ep_* values of 0.10 ± 0.002 and 0.21 ± 0.002 min^−1^, respectively. These habitats were labeled as “low-vascularity” habitats (LV). Altogether, the three identified habitats were as follows: high vascularity–high cellularity (HV-HC), low vascularity–high cellularity (LV-HC), and low vascularity–low cellularity (LV-LC).

### 3.2. Characterization of Tumor Imaging Phenotypes 

To elucidate intertumoral differences, we identified tumor imaging phenotypes using pre-treatment (day 0) habitat information for each tumor; that is, tumor imaging phenotypes describe subgroups of tumors distinguished by their microenvironment composition as defined by the fraction of each habitat prior to treatment (Figure 1). Variable tumor composition was observed at baseline across all tumors (Figure 3A), with a positive linear correlation found between the percent tumor volume of LV-LC and LV-HC habitats (*r* = 0.33, *p* < 0.01). Conversely, across all tumors a negative linear correlation was observed between the percent tumor volume of LV-HC and HV-HC habitats (*r* = −0.75, *p* < 0.01) and LV-LC and HV-HC habitats (*r* = −0.80, *p* < 0.01). Figure 3B shows the dendrogram resulting from the clustering analysis of baseline tumor data along with a heatmap displaying the percent tumor volume of each habitat for each tumor. The optimal number of clusters (i.e., phenotypes) was statistically determined to be two (Appendix A). The tumor imaging phenotypes were designated as Type 1 and Type 2, respectively.

Representative habitat maps of Type 1 and Type 2 tumors at day 0 are shown in Figure 3C. No statistically significant differences in volume were observed between phenotypes at baseline (*p* > 0.05). Type 1 tumors were comprised of a significantly higher fraction of HV-HC habitat compared to Type 2 tumors, with a median of 38.5% (18.0%) and 11.1% (10.2%) tumor volume, respectively (*p* < 0.01, Figure 3D). Type 2 tumors were comprised of significantly higher fractions of LV-HC and LV-LC habitats, with 44.6 % (12.8%) and 34.4% (6.2%), respectively, compared to Type 1 tumors, which demonstrated a distribution of 35.0% (24.8%) and 20.6% (9.8%), respectively (*p* < 0.01, Figure 3D). 

### 3.3. Quantifying Longitudinal Alterations in Tumor Volume

Figure 4A shows longitudinal changes in tumor volume for tumors treated with saline (control), trastuzumab, or paclitaxel over 30 days. At 30 days after the initiation of therapy, paclitaxel-treated tumors showed a significant decrease in median tumor volume of 19% (26%) compared to control tumors, which showed a 25% (69%) increase (*p* < 0.01). Tumors treated with trastuzumab showed a significant decrease in tumor volume compared to control and paclitaxel treated tumors, with a 78% (31%) decrease from baseline (*p* < 0.01). Similarly, tumors in the imaging cohort treated with trastuzumab showed a longitudinal decrease in tumor volume at day 4, with a median 6.8% (17%) decrease from day 0 (*p* = 0.01), significantly lower than control tumors, which showed a median 11% (22%) increase (Figure 4B). 

Figure 4C separates the tumors within each treatment group by tumor imaging phenotype. Type 1 control tumors showed a significant increase in tumor volume at day 1, with a median 8.4% (9.2%) increase compared to baseline (*p* < 0.05). Type 1 tumors treated with paclitaxel showed a decrease in tumor volume at day 1 of 7.1% (12%), trending towards significance (*p* = 0.06). Type 1 tumors treated with trastuzumab showed a significant decrease of 7.2% (15%) in tumor volume at day 4 compared to control, which had a 17% (15%) increase from baseline (*p* < 0.05). Compared to control tumors, Type 2 tumors showed no significant changes in tumor volume over the course of the MRI study. 

### 3.4. Quantifying Longitudinal Alterations in Tumor Composition

To evaluate changes in the tumor microenvironment, the temporal alterations in tumor composition as defined by the percent volume of each habitat were quantified. Figure 5 shows the longitudinal changes in tumor composition for Type 1 tumors in response to therapy. Type 1 control tumors showed no significant changes in tumor habitat composition over time (Figure 5A). Type 1 tumors treated with paclitaxel (Figure 5B) showed a significant decrease in the percent tumor volume of the HV-HC habitat, with a median proportion of 13.4% (6.9%) at day 4 compared to 40.4% (16.3%) at baseline (*p* < 0.01). A significant increase in the percent tumor volume of the LV-LC habitat was observed in Type 1 tumors treated with paclitaxel as well, with a median proportion of 45.6% (15.7%) at day 4 compared to 25.8% (7.4%) at day 0 (*p* < 0.01). Type 1 tumors treated with trastuzumab showed a significant increase in the percent tumor volume of LV-LC habitat, with a median proportion of 29.8% (10.4%) at day 4 compared to 21.1% (11.8%) at baseline (Figure 5C, *p* < 0.01).

Type 2 control tumors showed no significant changes in tumor habitat composition over time (Figure 6A). Type 2 tumors treated with paclitaxel (Figure 6B) showed a decrease in the percent tumor volume of the LV-HC habitat at day 1 (21.6% (8.6%)) compared to day 0 (46.0% (7.7%)), trending towards significance (*p* = 0.05). A corresponding increase (although not statistically significant) in the percent tumor volume of the HV-HC habitat was observed at day 1, with a median proportion of 46.8% (40.8%) compared to 11.0% (5.7%) at day 0. Type 2 tumors treated with trastuzumab (Figure 6C) showed a significant decrease in the percent tumor volume of the LV-HC habitat from baseline (45.7% (15.6%)), with a median of 22.5% (10.9%) at day 1 (*p* < 0.01) and 25.3% (12.2%) at day 4 (*p* < 0.05). A corresponding increase in the percent tumor volume of the HV-HC habitat (although not statistically significant) was observed at day 1, with a median proportion of 35.8% (43.5%) compared to 14.1% (14.9%) at day 0. Additionally, at day 1 Type 2 tumors treated with trastuzumab showed a significantly lower percent tumor volume in the LV-HC habitat compared to Type 1 tumors treated with trastuzumab (*p* < 0.01, Appendix A).

### 3.5. Correlation between MRI Habitats and Histology Data

Figure 7 shows the results of the immunofluorescence analysis of the excised tumors on day 5. Appendix A shows representative images of each immunofluorescence stain. Significant positive correlations were observed between the percent tumor volume of the HV-HC habitat and histological measures of vascularity, including the percent viable tissue area of CD31+ regions (*r*^2^ = 0.49, *p* = 0.03, Figure 7B) and the vascular maturation index (*r*^2^ = 0.57, *p* = 0.01, Figure 7D). Additionally, a significant positive correlation was observed between the percent tumor volume of the HV-HC habitat and the percent viable tissue area of the CD45+, F4/80+ region, staining for macrophage populations (*r*^2^ = 0.47, *p* = 0.04, Figure 7C). The LV-HC habitat showed a significant positive correlation with the percent viable tissue area of pimonidazole+ regions, indicating regions of hypoxia (*r*^2^ = 0.60, *p* < 0.01, Figure 7E). Appendix A shows the results of the immunohistochemistry analysis of the excised tumors on day 5. A negative correlation trending toward significance was observed between the percent tumor volume of the HV-HC habitat and the percent tissue area of necrosis identified from H&E-stained slides (*r*^2^ = −0.47, *p* = 0.05, Appendix A). No other significant correlations were observed between tumor habitats and the immunofluorescence or immunohistochemistry stains.

## 4. Discussion

Quantitative MRI habitats were used to identify tumor imaging phenotypes and quantify the longitudinal response to treatment for each individual phenotype in a xenograft model of HER2+ breast cancer. Quantitative parameters extracted from DW- and DCE-MRI allowed us to spatially resolve three tumor habitats, characterized in terms of cellularity or vascularity as HV-HC, LV-HC, and LV-LC. Tumor habitats were observed to spatially localize together, indicating that clustering of quantitative MRI parameters alone (i.e., without any spatial information) were capable of identifying physiologically-distinct tumor habitats. To further characterize the underlying biology of each habitat, we employed immunofluorescence and immunohistochemical staining of tumor tissues excised at the study endpoint. We observed significant correlations between the percent tumor volume of HV-HC habitats and histological measures of vascularity (CD31 and vascular maturation index) and macrophage infiltration (CD45+, F4/80+). These findings provide further evidence of the relationship between vasculature and macrophage infiltration observed in previous studies [29,30,31]; future work should explore habitat imaging in syngeneic mouse models with intact immune systems to further understand the relationship between habitats and immune infiltrates. Additionally, the LV-HC habitat, a subregion of increased cellular density and decreased vascular perfusion, was found to positively correlate with the percent area of pimonidazole-stained regions. This finding indicates that LV-HC regions may correspond to regions of hypoxia, providing a noninvasive MRI-derived measure of hypoxia.

Diverse microenvironmental compositions were observed at baseline across tumors (Figure 3A). Considering the correlation between vascular maturation and the HV-HC habitat, the negative correlation observed between the percent tumor volume of HV-HC and LV-LC habitats indicates that tumors with an increased percent tumor volume of HV-HC may have improved nutrient delivery and develop a lower fraction of LV-LC habitats. Conversely, the positive correlation between the percent tumor volume of LV-HC and LV-LC habitats suggests that the reverse may occur; i.e., regions of hypoxia or poorer nutrient delivery may lead to increased LV-LC subregions. 

Using tumor composition data, we identified two tumor phenotypes at baseline, designated as Type 1 and Type 2, and longitudinally evaluated each phenotype’s response to targeted and cytotoxic treatment. These imaging-defined tumor phenotypes were able to differentiate response to traditional targeted and chemotherapies, providing an early metric for response characterization. Type 1 tumors showed significantly increased percent tumor volume in the HV-HC habitat at baseline compared to Type 2 tumors, and without treatment (control) these tumors continued to increase in volume over the course of the MRI study. In previous work, both ourselves and others have found increased Ki-67 expression in tumor subregions associated with increased vascular perfusion [27,32,33]. Taken together, these findings suggest that an increased fraction of the HV-HC habitat may be associated with increased cell proliferation, contributing to continued tumor growth. Type 2 tumors were observed to have significantly higher fractions of the LV-HC and LV-LC habitats at baseline. These tumors exhibited no significant changes in tumor volume upon treatment with trastuzumab or paclitaxel, whereas longitudinal decreases in volume were observed in Type 1 tumors treated with the same therapies. These observations suggest that the Type 1 phenotype may be more sensitive to therapy compared to the Type 2 phenotype, potentially due to improved vascular delivery of therapeutics under normoxic conditions, which have been shown to increase sensitivity to treatment [34,35]. Furthermore, we observed therapy-specific changes in the tumor microenvironment indicative of response and distinct to each phenotype. Treated tumors of the Type 1 phenotype showed longitudinal increases in the LV-LC habitat (paclitaxel or trastuzumab) and decreases in the percent tumor volume of the HV-HC habitat (paclitaxel). Corresponding changes were not observed in treated tumors of the Type 2 phenotype; however, significant decreases in the LV-HC habitat were observed in trastuzumab-treated Type 2 tumors. These findings are consistent with previous observations of trastuzumab improving vascular perfusion [18,36] and tumor oxygenation [35,37] in preclinical models of HER2+ breast cancer, and demonstrate that alterations in the percent tumor volume of the LV-HC habitat may be an early indicator of trastuzumab response. 

Current approaches for characterization of clinical breast cancer involve molecular profiling of diagnostic biopsies, which are susceptible to sampling error due to intratumoral heterogeneity [7,13]. Other groups have investigated the utility of noninvasive imaging accompanied with radiomics techniques to quantify intertumoral differences and identify tumor phenotypes [38,39,40,41]. These approaches involve extraction and clustering of large feature sets (10 s to 100 s of features), and have been used to identify subgroups of patient tumors that correlate with, for example, breast cancer subtype [38,40] and recurrence-free survival [39,40]. While these approaches have demonstrated their prognostic value, the use of many complex features can make it difficult to interpret the intra- and intertumoral differences in tumor physiology. Although we used only three features to identify tumor phenotypes, quantitative MRI allows for interpretation of the underlying tumor microenvironment characteristics of each phenotype and the generation of hypotheses concerning the biological bases of therapeutic response. In the clinical setting, Wu et al. utilized tumor habitats extracted from DCE-MRI data to stratify breast cancer patients into two subgroups. They found that one subgroup was associated with increased risk of recurrence after neoadjuvant therapy, enabling prediction of recurrence-free survival from tumor habitat information. Their work, in addition to the clinical availability of quantitative DW- and DCE-MRI [42,43], demonstrates the potential for clinical translation of our tumor imaging phenotypes. The ability to resolve intertumoral differences and identify microenvironment characteristics related to therapy sensitivity could help guide treatment strategies for an individual patient.

There are four main areas for further investigation suggested by our study. First, we were not able to spatially register the histology and imaging data. While we did observe significant correlations between histological stains and habitats, lending biological support for tumor habitat physiologies, spatial registration of ex vivo histology data with in vivo MRI data would allow explicit linkage of spatial variations within a tumor to the underlying cellular diversity and tumor microenvironment. Additionally, further study is needed to understand the relationship between LV-LC habitats and necrosis. In the present study, we observed no significant correlation between the percent tumor volume of LV-LC and the percent tissue area of necrosis, potentially due to low variability in necrosis measures observed within 2D histology specimens (Appendix A). Second, the imaging parameter thresholds used to determine the habitat physiology (as high or low cellularity or vascularity) were selected manually. Future efforts to standardize methods of determining these thresholds are important for dissemination across a diverse setting of tumor types. Third, the stability of phenotype clusters (i.e., the consistency of clusters obtained from several data sets sampled from the same underlying distribution [44]) was not evaluated. Future work will investigate techniques that evaluate cluster stability to ensure repeatable identification of tumor imaging phenotypes. Finally, additional studies in orthotopic or syngeneic models of breast cancer are required to test the generalizability of the biological insights from this study. It is important to note, however, that the imaging and analysis methods described to resolve tumor heterogeneity are applicable to any tumor model.

## 5. Conclusions

We developed a novel spatially-resolved method for identifying and quantifying intratumoral heterogeneity using MRI habitats, then used this analysis to identify unique tumor phenotypes with differing responses to treatment in a preclinical model of HER2+ breast cancer. The Type 1 phenotype was associated with improved response to paclitaxel and trastuzumab. However, changes in tumor composition were observed in trastuzumab-treated Type 2 tumors, highlighting microenvironmental alterations indicative of therapeutic response. We provided further biological validation of the imaged habitats through correlation analysis using histological measures of vascular maturation and hypoxia. This methodology is capable of elucidating changes in spatiotemporal heterogeneity within tumors, and provides an approach to quantify intertumoral diversity. Application of these techniques in the clinical setting could improve understanding of an individual patient’s tumor pathology and potential therapeutic sensitivity.

## Figures and Tables

**Figure 2 cancers-14-01837-f002:**
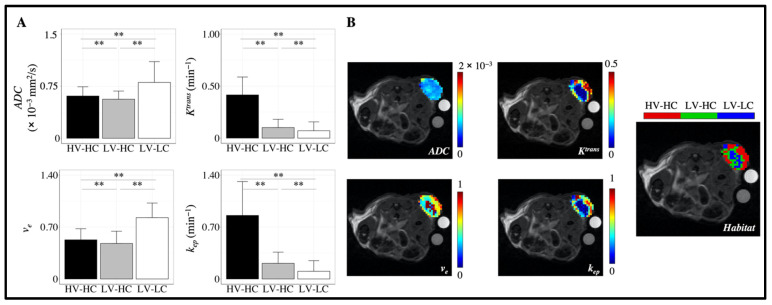
Characterization of tumor habitats. The mean value of each quantitative imaging parameter was used to determine the physiology represented by each habitat (**A**). Identified tumor habitats were labeled in terms of high or low “vascularity” (*K^trans^*, *k_ep_*) and “cellularity” (*ADC*, *v_e_*). Three tumor habitats were identified: high vascularity–high cellularity (HV-HC), low vascularity–high cellularity (LV-HC), and low vascularity–low cellularity (LV-LC). Error bars show standard deviation and ** indicates a *p* < 0.01. Representative parameter maps and corresponding habitat maps are shown in (**B**), with the HV-HC habitat shown in red, LV-HC in green, and LV-LC in blue. The units for *K^trans^* are mL (blood)/mL (tissue)/min, and the units for the *ADC* are mm^2^/s.

**Figure 3 cancers-14-01837-f003:**
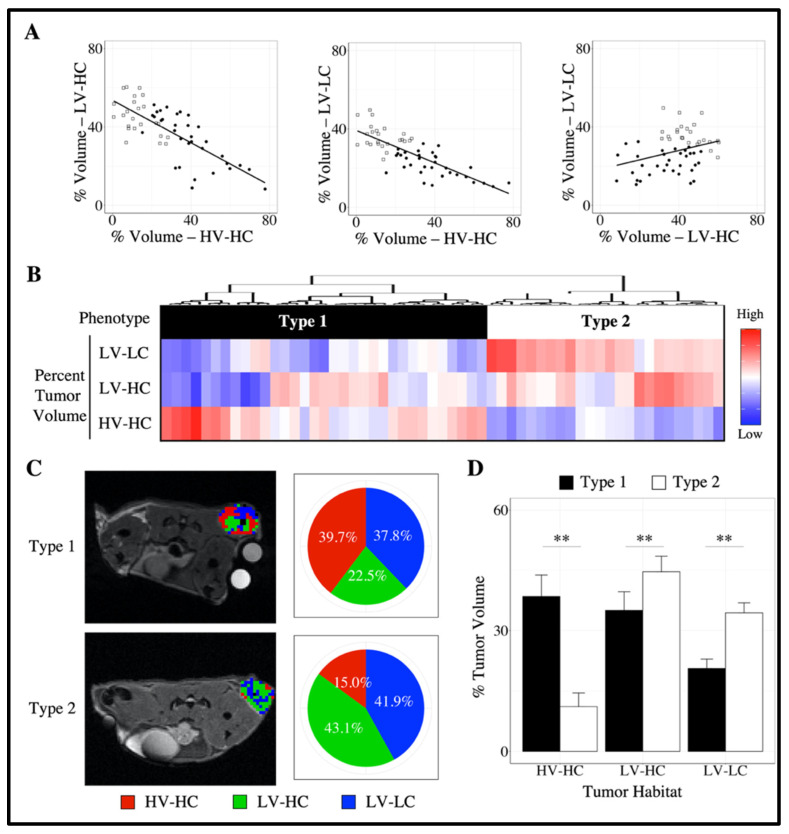
Discovery of tumor imaging phenotypes. (**A**) shows the linear relationship between tumor habitats at baseline. Each point represents a tumor at day 0 (pre-treatment), with tumor phenotypes represented by shape (Type 1: black circles, Type 2: white squares). A negative linear correlation was observed between the percent tumor volume of LV-HC and HV-HC habitats (*r* = −0.75, *p* < 0.01) and LV-LC and HV-HC habitats (*r* = −0.80, *p* < 0.01). A positive linear correlation was observed between the percent tumor volume of LV-LC and LV-HC habitats (*r* = 0.33, *p* < 0.01). (**B**) shows the resulting dendrogram and heatmap from agglomerative clustering of tumors (columns within the heatmap) using baseline tumor habitat information (rows), from which two phenotypes were identified. These tumor imaging phenotypes were designated as Type 1 (black) and Type 2 (white). (**C**) shows representative habitat maps of a Type 1 and Type 2 tumor at day 0 alongside a corresponding pie chart of whole tumor composition. Type 1 tumors showed significantly higher proportions of the HV-HC habitat compared to Type 2 tumors as well as decreased LV-HC and LV-LC habitats (**D**). Error bars show 95% confidence interval and ** indicates *p* < 0.01.

**Figure 4 cancers-14-01837-f004:**
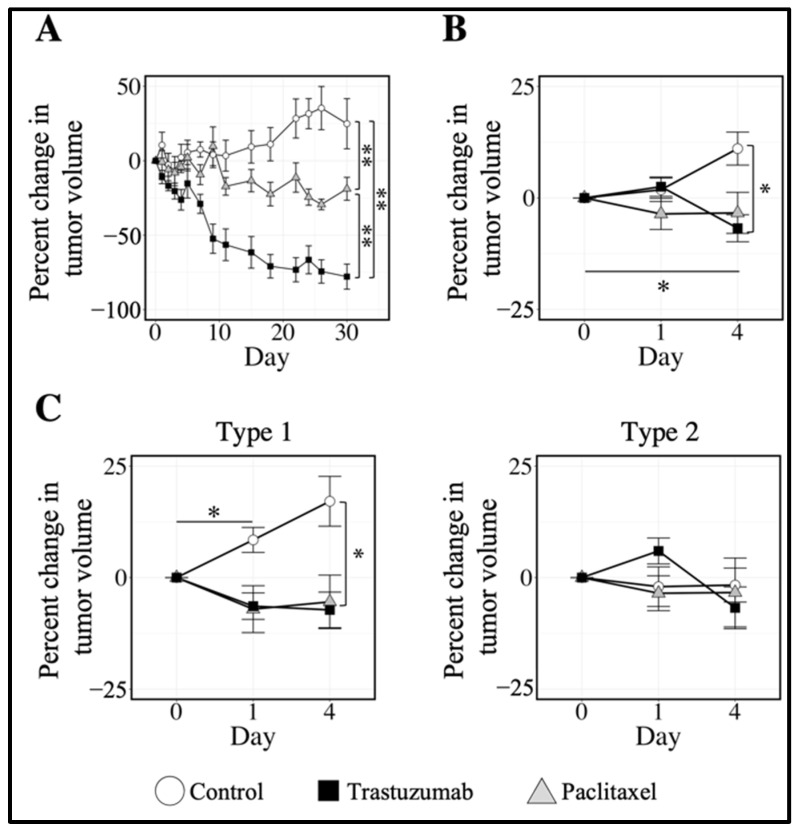
Longitudinal tumor response to targeted and cytotoxic therapies. (**A**) shows the median percent change in tumor volume over 30 days post-treatment for control, trastuzumab-treated, and paclitaxel-treated tumors. At 30 days after the initiation of therapy, tumors treated with paclitaxel showed a significant decrease in tumor growth compared to control tumors (*p* < 0.01). Treatment with trastuzumab yielded a longitudinal decrease in tumor volume, significantly lower than both control and paclitaxel-treated tumors (*p* < 0.01) at day 30. (**B**) shows the median percent change in tumor volume for control and trastuzumab- and paclitaxel-treated tumors over the course of the imaging study. Trastuzumab-treated tumors showed a longitudinal decrease in tumor volume at day 4 (*p* < 0.05) compared to baseline and significant decreases in tumor volume compared to control tumors at day 4 (*p* < 0.05). (**C**) shows the median percent change in tumor volume for each treatment group, as in (**B**), separated by tumor imaging phenotype. Type 1 control tumors showed a significant increase in tumor volume at day 1 (*p* < 0.05) compared to baseline. Type 1 tumors treated with trastuzumab showed a significant decrease in tumor volume compared to control tumors at day 4 (*p* < 0.05). Type 2 tumors showed no significant changes in tumor volume over the course of the MRI study. Error bars show standard error, ** indicates *p* < 0.01, * indicates *p* < 0.05.

**Figure 5 cancers-14-01837-f005:**
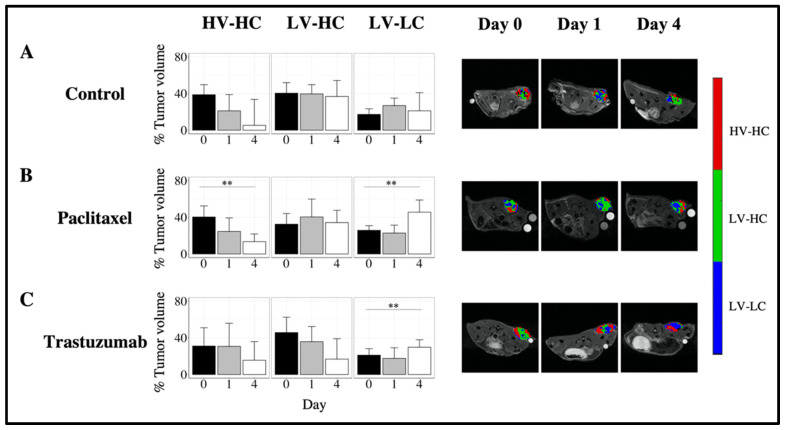
Longitudinal alterations in tumor composition of Type 1 tumors in response to cytotoxic and targeted therapies. The left column of each row (**A**–**C**) shows the median percent tumor volume of each habitat (HV-HC, LV-HC, LV-LC) for Type 1 tumors at days 0, 1, and 4. The right column of each row (**A**–**C**) shows representative habitat maps on days 0, 1, and 4 for Type 1 tumors. Longitudinal alterations in tumor composition are shown for control (row **A**), paclitaxel-treated (row **B**), and trastuzumab-treated (row **C**) tumors. Type 1 tumors treated with paclitaxel showed a longitudinal decrease in the percent tumor volume of the HV-HC habitat (*p* < 0.01) and a longitudinal increase in the percent tumor volume of the LV-LC habitat (*p* < 0.01). Type 1 tumors treated with trastuzumab showed a longitudinal increase in the percent tumor volume of the LV-LC habitat by day 4 (*p* < 0.05). No significant longitudinal alterations in tumor habitat composition were observed Type 1 control tumors. Error bars show standard deviation, ** indicates *p* < 0.01.

**Figure 6 cancers-14-01837-f006:**
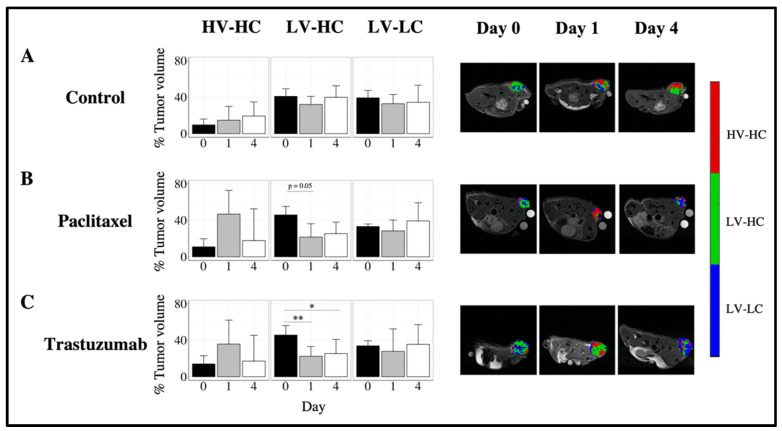
Longitudinal alterations in tumor composition of Type 2 tumors in response to cytotoxic and targeted therapies. The left column of each row (**A**–**C**) shows the median percent tumor volume of each habitat (HV-HC, LV-HC, LV-LC) for Type 2 tumors, at days 0, 1, and 4. The right column of each row (**A**–**C**) shows representative habitat maps at days 0, 1, and 4 for Type 2 tumors. Longitudinal alterations in tumor composition are shown for control (row **A**), paclitaxel-treated (row **B**), and trastuzumab-treated (row **C**) tumors. Type 2 tumors treated with trastuzumab showed a longitudinal decrease in the percent tumor volume of the LV-HC habitat at day 1 (*p* < 0.01) and day 4 (*p* < 0.05) compared to baseline. No significant longitudinal alterations in tumor habitat composition were observed Type 2 control tumors. Error bars show standard deviation, ** indicates *p* < 0.01, * indicates *p* < 0.05.

**Figure 7 cancers-14-01837-f007:**
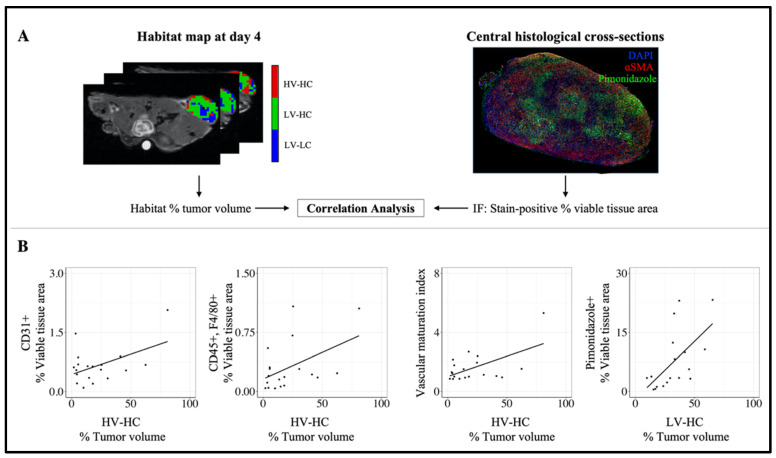
Correlation between MRI tumor habitats and immunofluorescence (IF) staining. For each tumor, the percent tumor volume of each habitat was calculated (**A**). For each tumor section and IF stain, stain-positive regions were quantified as the percent viable tissue area (**A**). Correlations were calculated between the percent tumor volume of each habitat and the percent viable tissue area of each IF stain. The percent tumor volume of the HV-HC habitat showed a significant positive correlation with the percent viable tissue area of the CD31+ (*r*^2^ = 0.49, *p* = 0.03) and CD45+, F4/80+ (*r*^2^ = 0.47, *p* = 0.04) regions (**B**, from left to right) as well as the vascular maturation index (*r*^2^ = 0.57, *p* = 0.01). The percent tumor volume of the LV-HC habitat showed a significant positive correlation with the percent viable tissue area of pimonidazole+ regions (*r*^2^ = 0.60, *p* < 0.01) (panel **B**, far right).

## Data Availability

The data presented in this study are available on request from the corresponding author.

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
