# Peer review of "Quantifying Tumor Heterogeneity via MRI Habitats to Characterize Microenvironmental Alterations in HER2+ Breast Cancer"

_cancers, 2022, doi:10.3390/cancers14071837_

Round 1

Reviewer 1 Report

In this article, the authors used magnetic resonance imaging to define the importance of the microenvironment in the response to treatments. They demonstrate by image analysis and tumor parameters that there is intra-tumor heterogeneity. This heterogeneity conditions the response to treatments. It is expected that the most vascularized tumors have a higher sensitivity to treatments than the non-vascularized ones.

The experiments appear to be well conducted and the results support the conclusions

Still, I have a few questions of curiosity that the authors could answer.

  • In nude mice, macrophages are more numerous and have a stronger phagocytic activity. How do the authors fit these data into their conclusions?
  • An example of some images obtained in ICC for each of the markings should be attached as additional data.
  • we do not distinguish the cells on the photos in additional data is that the authors can enlarge the marked area to see the exact location of the marking.
  • It would be nice to update the bibliography with some recent journals and add those in the introduction or discussion. “MRI-based machine learning radiomics can predict HER2 expression level and pathologic response after neoadjuvant therapy in HER2 overexpressing breast cancer “

Author Response

Dear Editor and Reviewers,

Thank you for the opportunity to submit a revised manuscript and for the time you have committed to improving the study. We appreciate your efforts and believe we have addressed all of your points as indicated below and in the revised manuscript.

Reviewer 1

Comments to the Author

In this article, the authors used magnetic resonance imaging to define the importance of the microenvironment in the response to treatments. They demonstrate by image analysis and tumor parameters that there is intra-tumor heterogeneity. This heterogeneity conditions the response to treatments. It is expected that the most vascularized tumors have a higher sensitivity to treatments than the non-vascularized ones. The experiments appear to be well conducted and the results support the conclusions. Still, I have a few questions of curiosity that the authors could answer.

  1. In nude mice, macrophages are more numerous and have a stronger phagocytic activity. How do the authors fit these data into their conclusions?

Response: Thank you for your comments and favorable review of our manuscript. In this study, we observed a significant positive correlation between percent tumor volume of the high vascularity - high cellularity (HV-HC) habitat and macrophage infiltration, measured as percent viable tissue area of CD45+, F4/80+ regions. Indeed, nude mice have been shown to have elevated macrophage activity. While both chemotherapy and trastuzumab targeted therapy have been shown increase phagocytic activity in tumors [1–3], in this study we observed no significant differences in macrophage levels between different treatment and control groups, as measured by immunofluorescent staining. The mechanisms of immunogenicity are multifaceted and the timing window of our investigations may not provide differences these differences. Future work will explore habitat imaging in syngeneic mouse models with intact immune systems, to better understand the relationship between tumor habitats and macrophage infiltration (as well as other adaptive immune responses and immune infiltration such as T-cells). We have expanded upon our future directions is in the discussion as follows:

“…future work should explore habitat imaging in syngeneic mouse models with intact immune systems to further understand the relationship between habitats and immune infiltrates.”

  1. An example of some images obtained in ICC for each of the markings should be attached as additional data.

Response: Thank you for this suggestion, we have added representative immunofluorescence images as Figure S5 in the supplemental materials.

  1. We do not distinguish the cells on the photos in additional data is that the authors can enlarge the marked area to see the exact location of the marking.

Response: We are not certain what the reviewer is referring to, but we think the reviewer is requesting an image that shows individual cells in the IHC images.  While such data were not central to this study, we have included zoomed in regions in Figure S5 so the reviewers can visualize stains at the cellular level.

  1. It would be nice to update the bibliography with some recent journals and add those in the introduction or discussion. “MRI-based machine learning radiomics can predict HER2 expression level and pathologic response after neoadjuvant therapy in HER2 overexpressing breast cancer”

Response: Thank you for this suggestion, we have included this reference and other more recent work in the introduction and discussion sections.

  1. Feng, M.; Jiang, W.; Kim, B.Y.; Zhang, C.; Fu, Y.-X.; Weissman, I.L. Phagocytosis Checkpoints as New Targets for Cancer Immunotherapy. Nat. Rev. Cancer 2019, 19, 568–586, doi:10.1038/s41568-019-0183-z.
  2. Lecoultre, M.; Dutoit, V.; Walker, P.R. Phagocytic Function of Tumor-Associated Macrophages as a Key Determinant of Tumor Progression Control: A Review. J. Immunother. Cancer 2020, 8, e001408, doi:10.1136/jitc-2020-001408.
  3. Chen, S.; Lai, S.W.T.; Brown, C.E.; Feng, M. Harnessing and Enhancing Macrophage Phagocytosis for Cancer Therapy. Front. Immunol. 2021, 12.

Reviewer 2 Report

The authors studied well-equipped imaging techniques and especially the MRI to accurately investigate tumor heterogeneity. The authors have blended the current knowledge and critical advances needed to investigate and quantify tumor heterogeneity advance methods, which are quite worthful. However, a few queries related to the paper would be like;

  1. What do the authors think on further delineating necrosis and Ki-67 expression levels for understanding the physiologies of the identified histological habitats.
  2. By integrating histograms into this study, the authors need to explain how tumor heterogeneity through MRI is demonstrated?
  3. The authors should also consider giving a more detailed discussion of their findings through figure S2.
  4. I suggest strengthening the introduction with relevant studies to clarify the study's objectives much more clearly.
  5. There are many typo errors that need to be polished by a native English speaker in the manuscript.

Author Response

Dear Editor and Reviewers,

Thank you for the opportunity to submit a revised manuscript and for the time you have committed to improving the study. We appreciate your efforts and believe we have addressed all of your points as indicated below and in the revised manuscript.

Reviewer 2

Comments to the Author

The authors studied well-equipped imaging techniques and especially the MRI to accurately investigate tumor heterogeneity. The authors have blended the current knowledge and critical advances needed to investigate and quantify tumor heterogeneity advance methods, which are quite worthful. However, a few queries related to the paper would be like;

  1. What do the authors think on further delineating necrosis and Ki-67 expression levels for understanding the physiologies of the identified histological habitats.

Response: Thank you for this comment. In previous work, we explored Ki-67 expression and necrosis and found significantly increased Ki-67 expression in histological high vascularity – high cellularity (HV-HC) habitats [1], as well as significantly increased necrosis in histological low vascularity - low cellularity habitats. Additionally, we found significant correlations between histological and MRI tumor habitats, suggesting that lesions with increased HV-HC in vivo may have increased Ki-67 expression. In future work, we plan to explore histopathological methods that allow for spatial correlation of in vivo habitats and ex vivo histological measures.

  1. By integrating histograms into this study, the authors need to explain how tumor heterogeneity through MRI is demonstrated?

Response: We appreciate the reviewer’s interest in our work. In this study, we used multiparametric MRI to delineate tumor habitats, breaking a tumor down into component subregions. We are unclear to which histograms the reviewer is referring to, as no histogram analyses were performed in this study. Previous work demonstrated the use of histograms to explore tumor heterogeneity [2] in response to targeted therapy.

  1. The authors should also consider giving a more detailed discussion of their findings through figure S2.

Response: Thank you for this suggestion. In certain lesions, we observed susceptibility artifacts that led to signal void regions across T1-weighted,  T2-weighted,  and diffusion weighted images. As these areas of the MRI did not contain any signal, we excluded it from subsequent analysis. We hypothesized the source of these susceptibility artifacts were due to calcifications. H&E staining of the same lesions was performed, and in some cases cross-sections showed calcifications as indicated in Figure S2. Calcifications have been previous demonstrated to have low signal in MRI [3]. We have included these points in the revised supplemental methods.

  1. I suggest strengthening the introduction with relevant studies to clarify the study's objectives much more clearly. 

Response: Thank you for this suggestion, we have updated the references in the revised Introduction and Discussion sections.

  1. There are many typo errors that need to be polished by a native English speaker in the manuscript.

Response: We thank the reviewer for their careful assessment of our manuscript. We have reviewed the manuscript and were unable to identify any typographical errors or grammatical issues. If the revised manuscript still contains errors, we will happily correct any identified errors.

References:

  1. Syed, A.K.; Whisenant, J.G.; Barnes, S.L.; Sorace, A.G.; Yankeelov, T.E. Multiparametric Analysis of Longitudinal Quantitative MRI Data to Identify Distinct Tumor Habitats in Preclinical Models of Breast Cancer. Cancers 2020, 12, 1682, doi:10.3390/cancers12061682.
  2. Syed, A.K.; Woodall, R.; Whisenant, J.G.; Yankeelov, T.E.; Sorace, A.G. Characterizing Trastuzumab-Induced Alterations in Intratumoral Heterogeneity with Quantitative Imaging and Immunohistochemistry in HER2+ Breast Cancer. Neoplasia 2019, 21, 17–29, doi:10.1016/j.neo.2018.10.008.
  3. Wehrli, F.W. Magnetic Resonance of Calcified Tissues. J. Magn. Reson. 2013, 229, 35–48, doi:10.1016/j.jmr.2012.12.011.

Reviewer 3 Report

Immunofluorescence photos of the following antibodies must be provided: anti-CD31 (R&D Systems, Minneapolis, MN), anti-pimonidazole (Hypoxyprobe, Inc., Burlington, MA), anti-CD45, or anti-F4/80 (Abcam, Cambridge, UK).

Author Response

Dear Editor and Reviewers,

Thank you for the opportunity to submit a revised manuscript and for the time you have committed to improving the study. We appreciate your efforts and believe we have addressed all of your points as indicated below and in the revised manuscript.

Reviewer 3

Comments to the Author

  1. Immunofluorescence photos of the following antibodies must be provided: anti-CD31 (R&D Systems, Minneapolis, MN), anti-pimonidazole (Hypoxyprobe, Inc., Burlington, MA), anti-CD45, or anti-F4/80 (Abcam, Cambridge, UK).

Response: Thank you for this suggestion, we have added representative immunofluorescence images as Figure S5 in the supplemental materials.

Round 2

Reviewer 3 Report

1 L356, Figure S6 shows the results of immunohistochemistry analysis of excised tumors on day 5.  It is Figure S5 in L356.
2 Figure S5 missing magnification.

Author Response

We thank the reviewer for their comments. We have amended Figure S5 to include scale bars to indicate the magnification of the images.

Additionally, Figure S6 shows results of our immunohistochemistry analysis. We have added a sentence in the main text explicitly referencing the immunofluorescent images in Figures S5 to reduce confusion.